# From Burst to Sustained Release: The Effect of Antibiotic Structure Incorporated into Chitosan-Based Films

**DOI:** 10.3390/antibiotics13111055

**Published:** 2024-11-06

**Authors:** Nathália F. Sczesny, Helton J. Wiggers, Cecilia Z. Bueno, Pascale Chevallier, Francesco Copes, Diego Mantovani

**Affiliations:** 1Laboratory for Biomaterials and Bioengineering (LBB-BPK), Associação de Ensino, Pesquisa e Extensão BIOPARK, Max Planck Avenue, 3797, Building Charles Darwin, Toledo 85919-899, PR, Brazil; nsczesny@outlook.com (N.F.S.); helton.wiggers@bpkedu.com.br (H.J.W.); cecilia.bueno@bpkedu.com.br (C.Z.B.); 2Laboratory for Biomaterials and Bioengineering (LBB-UL), Canada Research Chair Tier I, Department of Min-Met-Materials Engineering & Division Regenerative Medicine of CHU de Quebec Research Center, Laval University, Quebec City, QC G1V 0A6, Canada; francesco.copes.1@ulaval.ca

**Keywords:** antibacterial film, sustained drug delivery, medical devices, structure-property relationships

## Abstract

**Background/Objectives:** Medical devices are susceptible to bacterial colonization and biofilm formation, which can result in severe infections, leading to prolonged hospital stays and increased burden on society. Antibacterial films have the potential to assist in preventing biofilm formation, thereby reducing administration of antibiotics and the emergence of antibiotic-resistant strains. In a previous study, a chitosan-based matrix crosslinked with tannic acid and loaded with gentamicin was reported. In this study, five different antibiotics (moxifloxacin, ciprofloxacin, trimethoprim, sulfamethoxazole or linezolid) were loaded into these chitosan-based films, and their impact on the release behavior carefully assessed. **Methods:** The samples were characterized according to their thickness, swelling, and mass loss in phosphate-buffered saline (PBS), as well as by morphology using scanning electron microscopy (SEM) and optical phase contrast microscopy. Antibiotic release over time was quantified in PBS by high-performance liquid chromatography (HPLC). Antibacterial activity was investigated by disk diffusion test and antibiotic release over time. Finally, the cytotoxicity of the samples was assessed with human dermal fibroblasts. **Results:** The obtained results differed significantly, especially regarding the antibiotic release time and antibacterial activity, which varied from one day to six months, enabling classification of the films from burst/transient to prolonged release. The films also showed antibacterial features against bacteria mostly present in medical devices and displayed to be non-cytotoxic. **Conclusions:** In conclusion, it was demonstrated that the antibiotics structure significantly alters the release kinetics, and that by carefully selecting the antibiotic, the consequent release can be tuned. This approach yielded films that could be used for potentially-scalable release in antimicrobial coatings specific to medical devices, aiming to reduce biomaterial associated infections (BAIs).

## 1. Introduction

The World Health Organization (WHO) reports that medical devices, such as central venous catheters (CVCs), urinary catheters, and ventilators, are essential for the treatment of patients in different clinical settings [1]. However, they are highly susceptible to Healthcare-Associated Infections (HAIs). These infections are most often caused by coagulase negative Staphylococci, such as *Staphylococcus aureus*, *Klebsiella pneumoniae*, *Pseudomonas aeruginosa*, *Enterococcus faecalis* and *Candida* species [2,3]. To prevent these infections, primary protocols are recommended, including hand hygiene, maintaining a hygienic hospital environment, patient categorization, public health surveillance, and oral administration of antibiotics [4]. Although prevention is one of the most important actions, it is insufficient: The Centers for Disease Control and Prevention (CDC) in the USA recently reported that 1 in 31 hospital patients per day acquires a HAI [5]. The most frequently reported HAIs involve central–line-associated bloodstream infections (CLABSI), catheter-associated urinary tract infections (CAUTI) and ventilator-associated pneumonia (VAP) [6]. It is, therefore, necessary to develop technologies that make medical devices antibacterial.

Among the different strategies that have been developed, the deposition of coatings able to prevent biofouling and/or kill bacteria on contact are of great interest. These coatings may contain active ingredients such as antibiotics, metal ions, nanoparticles, or compounds released by contact or activated by light [7]. The main requirements for these systems are stability, biocompatibility and antibacterial activity for the entire time that the medical device is in contact with the patient [8,9].

The U.S. Food and Drug Administration reports that medical devices have dwell times ranging from a few hours to over 30 days. This period can be classified as limited exposure (or transient) if used for less than 60 min, prolonged exposure (or short-term) if used between 60 min and 30 days, and permanent exposure (or long-term) if used for more than 30 days [10].

Central venous catheters (CVC), for example, are classified according to their dwell time, which can range from days (peripherally inserted central catheter, or PICC) to years (port-a-cath) depending on the device. Several studies have revealed a temporal relationship between CVC dwell time and catheter colonization by microorganisms, i.e., the longer the time, the higher the incidence of infection. Therefore, longer implantation represents higher risk [11]. However, the catheters currently available on the market, which contain impregnated antimicrobial compounds, have a release time of less than 30 days (Table 1).

Given the varying indwelling times associated with different CVCs, it is imperative to design a system with drug delivery kinetics that aligns with the specific needs of each catheter. Hence, antibacterial coatings with tunable releasing time offer a promising strategy to improve the use of CVC catheters [16]. In order to design an antibacterial coating, biopolymers have been considered, mainly due to their biodegradable, biocompatible, and non-toxic properties [17].

Among the different biopolymers, chitosan, an amino polysaccharide derived from chitin, which is present in different living beings, was selected as a promising matrix. Additionally, it is biocompatible, cost-effective and environmentally-friendly [18,19]. By adding a crosslinker, the mechanical and drug-release properties of the chitosan-based material can be modulated. Moreover, chitosan has intrinsic antibacterial properties [20,21] and, together with the loaded antibiotic, it can have a dual antibacterial effect: contact killing and drug releasing. Furthermore, in some cases, synergic effects with antibiotics were observed [22], where antibiotic potency increased with combination. However, in this work, the focus was on antibiotic release, and synergistic effects were not evaluated. A previous publication demonstrated the effectiveness of tannic acid (TA) with iron sulphate (FeSO_4_) as a chitosan crosslinker in films loaded with gentamicin, enabling a prolonged release of the antibiotic up to 35 days [23].

Despite several studies reporting the use of crosslinkers to control drug release [24,25,26,27], the influence of the drug’s structure in the release kinetics is still, to the best of our knowledge, unexplored. Drug structures and functional groups are expected to play an important role in the release, since they influence the strength of the interactions with the polymer matrix [28,29,30].

In this study, five different antibiotics were selected based on: molecular weight, to investigate the effect of size on release kinetics; solubility, to observe any trends related to low solubility on slow release; and diversity of functional group and electrostatic charge, to identify any effects of the intermolecular interaction between the antibiotics and the film matrix (polymer and crosslinkers interactions), as shown in Table 2.

## 2. Results and Discussions

### 2.1. Antibiotic Release over Time

The cumulative release of the five antibiotics from the films is shown in Figure 1. Sulfamethoxazole, which has a negative charge, and linezolid, which has a neutral charge, were both released from the films in 1 day, being considered as transient release. On the other hand, positively charged antibiotic trimethoprim was released in 28 days, characterizing a short-term release. Finally, the two zwitterionic antibiotics, ciprofloxacin and moxifloxacin, were released in 120 and 180 days, respectively, being considered as long-term release.

Moxifloxacin presented the longest release time, more than 180 days, which is remarkably longer than what is normally reported in the literature for drug release from chitosan-based matrices. For instance, chitosan/silver nanocomposite films loaded with moxifloxacin released the drug for up to 48 h [31]; chitosan hydrogels with embedded microgels containing moxifloxacin displayed pH and temperature-dependent release, which was sustained for 60 h [32]; chitosan/alginate-based multilayers sustained the release of moxifloxacin for almost 180 h [33]; chitosan-based coatings crosslinked with caffeic acid exhibited sustained moxifloxacin release for up to 49 days [34]. To the best of our knowledge, the literature has reported only one system with ultra-long-term drug release, consisting of a PLGA-based matrix loaded with MK-2048 HIV antiviral drug, which was released up to 200 days [35].

In the first 24 h of the experiment, less than 20% of the loaded antibiotics ciprofloxacin and moxifloxacin were released, while the others exceeded 40%, therefore, burst release was reduced by at least two times. Moreover, MOX and CIP exhibited the most controlled release profiles, with a gradual increase over time.

It was observed that antibiotic molecular weight had no influence on the release kinetics. Although trimethoprim and sulfamethoxazole have the lowest molecular weights, they were classified in different release time categories, i.e., trimethoprim was considered to have short-term release and sulfamethoxazole, transient release. Moreover, when comparing ciprofloxacin and linezolid, which have similar molecular weights, their release times were very different, 120 days for ciprofloxacin and 1 day for linezolid. Additionally, information regarding partition coefficient octanol/water, topographic polar surface area, number of rotatable bonds (which relation to the structural properties of the antibiotics) shows no clear trends regarding release (see Appendix A). Therefore, the data lead to the conclusion that the molecular interactions between the antibiotics and the components of the films are more important than their molecular weight and structural properties to determine the release rate.

The crosslinker and polymer used are also expected to play important roles in drug release kinetics. However, since in this study these parameters were fixed, the differences from transient to prolonged antibiotics release can be attributed to different interactions of the drugs with the chitosan-based film matrix [36,37].

According to the structures shown in Figure 1C, linezolid has no ionizable functional group at physiological pH of 7.4. Sulfamethoxazole has a sulfonamide functional group, that can be negatively charged at pH 7.4; however, there is a high delocalization of the charge [38]. The interactions of linezolid and sulfamethoxazole with the chitosan-based film may be driven by hydrogen bonding. On the other hand, trimethoprim, ciprofloxacin and moxifloxacin possess positive charges at pH 7.4 with the possibility of charge–charge interactions with the film matrix, more specifically with the tannic acid galloyl moieties.

Since charge–charge interactions are stronger than hydrogen bonds, they can be associated with slower and more controlled release kinetics. Therefore, the complementary intermolecular interactions between the fluoroquinolones and the film matrix are responsible for the significant increase in release time [39,40] compared to the other antibiotics tested in this study. Figure 2 illustrates the hypothesis.

It is also possible to notice that the antibiotics exhibit different release percentages, which may be due to different factors or their combinations, such as antibiotic electric charge, which influences the intermolecular interactions, and water solubility, which influences the distribution of the antibiotic molecules in the polymeric film. These main factors are summarized in Table 2. For instance, it can be seen that antibiotics with lower solubility (MOX and SMX) were released at higher percentages, while the antibiotics with higher solubility (TMP, CIP and LNZ) were released at lower percentages (Figure 1). This may be explained by the fact that the less water-soluble antibiotics are not well dispersed in the polymer matrix, and tend to be distributed over the polymer surface [41]. The difference between MOX and SMX is that MOX interacts strongly with the polymer matrix, thus delaying its release. TMP, CIP and LNZ, on the other hand, are more water-soluble and tend to disperse in the polymer matrix, so that the release of all incorporated antibiotics remains at a maximum of 50%.

The different interactions among the molecules in the matrix can influence the mechanisms that govern release. There are three main categories of drug release mechanisms: swelling, which increases the space between polymeric chains; erosion, which promotes surface degradation; and diffusion, when drug molecules move according to their concentration gradient from a higher to a lower concentration [42]. Moreover, a system can be governed by a combination of mechanisms.

In order to ascertain which mechanism is responsible for the release of antibiotics from chitosan-based films, a variety of mathematical models were used to fit to the release data. The data collected up to 180 min (as described in Section 3.3) were considered for the fitting, taking into account a maximum of 60% of the cumulative drug release, in accordance with the recommendations of Peppas-Sahlin [43]. Table 3 presents the R^2^ values for each fitting and their respective mechanisms. The Model Selection Criterion (MSC) and Akaike Information Criterion (AIC) values serve as supplementary parameters in the process of identifying the optimal fitting; in comparison to other models, the best fitting is the one with the largest MSC and the lower AIC (Appendix A). These parameters have been used by different authors in terms of dissolution data modelling [44,45].

The highest goodness of fit (R^2^) and MSC and lowest AIC values (see Appendix A) were observed for the Korsmeyer-Peppas and Weibull models, regardless of the antibiotic loaded into the films. The fact that different models exhibited good fitting may be indicative of a complex and mixed mechanism. However, it was also observed that MOX, CIP, TMP, and LNZ exhibited good fits to the Higuchi model, which suggests that diffusion may be the predominant mechanism.

The fittings for the Korsmeyer-Peppas and Weibull mathematical models were interpreted, presented in Table 4 [43,46,47]).

It is noteworthy that the different release mechanisms of the antibiotics align with long-term, short-term, and transient release profiles, as showed in Table 4. For MOX and CIP, the kinetic parameters indicated an anomalous or non-Fickian mechanism for Korsmeyer-Peppas and diffusion with an additional mechanism for Weibull. SMX and LNZ are released by Fickian diffusion according to Korsmeyer-Peppas and diffusion through high disordered spaces for Weibull, which may describe a percolation by preferential ways. TMP is classified as Fickian by both models.

The differences observed in the release kinetics and mechanism are indicative that, besides the interactions of the drugs with the matrix, the homogeneity and physicochemical characteristics of the polymer matrix may impact the release behavior. As pointed out in the literature, the factors that affect drug release rate are both polymer and drug chemical properties and the structure of the polymer matrix. Consequently, factors such as porosity, homogeneity and crystallinity are also determinant for the release profile [48]. The physicochemical characterization of the films was therefore explored to support these observations.

To better understand the impact of the film’s morphology on the release kinetics, visual inspection, optical microscopy and scanning electron microscopy (SEM) analyses were carried out. The results are presented in Figure 3. The samples selected for this analysis were the films loaded with moxifloxacin and sulfamethoxazole, which had the most significant differences in release time, as well as a blank film without antibiotics.

Visual inspection did not show significant differences among the films. SEM analysis also indicated similarities in the film surfaces, which presented a dense and compact structure that is commonly noticed with chitosan films [49,50,51]. Optical microscopy revealed inner details of the film, indicating a significant heterogeneity in the morphology of SMX films, while MOX and blank films showed good homogeneity. The noticeable grooves in SMX may be formed during the casting process. Since the methodology used to prepare the films was the same, this phenomenon can be explained by the difference in solubility between SMX and MOX [52,53], and intermolecular interaction between the antibiotics and polymer matrix within the films.

Since the images obtained by SEM did not reveal any discernible differences in the surfaces of the films, it is possible to assume that the observed heterogeneity of SMX films is located within its structure, with irregularly distributed pores. According to the literature, this porous structure may increase burst release [54], which corroborates the obtained results shown in Figure 1. On the other hand, moxifloxacin is released mainly by diffusion, in a more controlled manner, probably because of the high homogeneity of the film.

### 2.2. Physicochemical Characterization

The films were characterized according to their thickness and swelling in PBS, as shown in Table 5.

A film without antibiotic (blank) was used for comparison with the films containing different antibiotics. There was no increase in the thickness of the antibiotics-loaded films compared to the blank, supporting that the drugs were incorporated into the interstitial spaces of the film matrix. The blank film showed higher swelling compared to the antibiotic-containing films. Swelling in chitosan films can be influenced by the crosslinking density [55,56]. Also, drug-polymer interactions may physically crosslink the film, especially due to charge–charge and hydrogen bonding interactions, as shown in Figure 2. Although the swelling for SMX and LNZ films is slightly higher, there were no statistically significant differences (*p* < 0.05) among the films prepared with different antibiotics. In addition, no trend was observed for mass loss, showing similar degradation for all films. These results suggest that the presence of antibiotics does not significantly affect the overall swelling and degradation behavior of the chitosan matrix.

Regarding release kinetics, it can be inferred that film thickness and swelling in aqueous fluids is not correlated with the significant difference in the antibiotic release mechanisms. This corroborates the analysis of release mechanism performed in Section 2.1, which was not shown to be controlled by swelling.

### 2.3. Antibacterial Activity

In order to verify the efficacy of the films as an antibacterial material, disk diffusion analysis was performed against six bacteria commonly found in Healthcare Associated Infections (HAI). The positive controls were prepared according to standardized method [57] and compared to the Eucast Database [58,59], and the results are presented in Appendix A. Then, the films containing the antibiotics were tested; the results are shown in Table 6.

The disk diffusion showed that the films incorporated with the antibiotics possesses strong antibacterial properties. Considering that all the antibiotics were completely incorporated after the casting, the dose of antibiotics in the 6 mm disc was 22 µg, which is higher than the dose for standard sensibility disk diffusion tests [60], and therefore antibacterial activity is expected upon antibiotic release. Films loaded with ciprofloxacin and moxifloxacin, which are both broad-spectrum antibiotics, demonstrated activity against all bacteria tested, with inhibition halo varying from 19.4 to 33.7 mm. In contrast, films loaded with linezolid, trimethoprim and sulfamethoxazole, narrow-spectrum antibiotics, did not demonstrate activity against all bacteria. TMP film demonstrated no activity against *P. aeruginosa*, SMX film exhibited no activity against *S. aureus*, *S. epidermidis*, *P. aeruginosa* and *E. faecalis* and LNZ film exhibited no activity against *E. coli* and *P. aeruginosa*. These results were expected, since linezolid acts against gram-positive bacteria and, with rare exceptions, against gram-negative bacteria [61,62]. Moreover, trimethoprim and sulfamethoxazole did not present good efficacies when studied separately, since they are commonly used together for synergistic effects [63].

In addition, an antibiotic minimum inhibitory concentration (MIC) assay was performed to validate and standardize the methodology used in this study, the results of which are presented in Appendix A. The concentration of antibiotics released from the films over time must be greater than the MIC concentration to promote antibacterial activity. Figure 4 displays the antibiotics concentrations measured in PBS and the MIC range for the bacteria frequently found in medical device infections *E. coli*, *P. aeruginosa*, *S. aureus*, *S. epidermidis*, *K. Pneumoniae* and *E. faecalis*.

According to Figure 4, it is expected that the released antibiotics can prevent bacterial infection when their concentrations are above the MIC range. For example, CIP should prevent bacterial contamination at least for 80 days, MOX for 150 days, TMP for 20 days and SMX and LNZ for 1 day.

To confirm this, indirect antibacterial activity tests over time were carried out against two representative microorganisms *S. aureus* (gram positive) and *E. coli* (gram negative). Figure 5 presents the percentage of bacterial survival following contact with the aliquots collected in the antibiotic release experiments in culture media (item 3.6.3). The results showed contamination of LNZ and SMX films on the first day for both *E. coli* and *S. aureus*, corroborating the release experiments in PBS (Figure 1), and supporting the classification of SMX and LNZ as transient release systems. TMP films showed antibacterial activity up to 60 days for *E. coli* and 21 days for *S. aureus*, corroborating its previous classification as a short-term release system. Finally, MOX and CIP did not present any contamination until six months for any bacterium, and thus can be considered long-term release systems.

The analysis of the indirect antibacterial activity over time corroborates the analysis of the release kinetics in PBS in terms of temporal classification. However, some differences can be noticed when comparing the two experiments. This discrepancy is likely due to the significant differences in the composition of the media where the antibiotics were released. The phosphate-buffered saline (PBS) solution, employed for release kinetics, is a salt solution with a much less complex composition than the Mueller Hinton broth, which is composed of amino acids, nitrogenous substances, and other nutrients [64]. Consequently, the antibiotics diffusion is slower in Mueller Hinton broth compared to PBS media, resulting in a lower release rate.

### 2.4. Indirect Cytotoxicity Test

Figure 6 shows the results of the performed indirect cytotoxicity test. As can be seen, the extracts obtained from TMP and MOX caused a reduction of cell viability (less than 90%) compared to the CTRL (*p* ˂ 0.05). All the other tested conditions did not exert negative effects on the treated cells. According to ISO 10993-5 [65], the accepted rate for the cytotoxicity assay is above 80%, and there were no conditions under this limit. This means that in terms of in vivo application, all the films are likely to be biocompatible, and it is highly motivating to continue the investigation on animals in future work.

As previously stated, chitosan is biocompatible, and Rodrigues et al. confirms that a concentration lower than or equal to 1 mg/mL is not cytotoxic [66]. On the other hand, tannic acid, which acts as a crosslinker, is described as potentially cytotoxic [66], as are antibiotics, depending on the concentrations. Consequently, the slight reduction in cell viability observed for MOX and TMP can be attributed to both tannic acid or the antibiotic.

## 3. Materials and Methods

### 3.1. Materials

Chitosan (Sigma, medium molecular weight, Shanghai, China), tannic acid (ACS reagent, Sigma, Pequim, China), phosphate-buffered saline (PBS) (Sigma, Gillingham, UK), iron sulphate heptahydrate (Êxodo Científica, Sumaré, Brazil), moxifloxacin hydrochloride (Prati Donaduzzi, Toledo, Brazil), trimethoprim (Haishing CO PTE. Ltd., Singapore), linezolid (Prati Donaduzzi, Toledo, Brazil), sulfamethoxazole (Virchow Laboratories Limited, Telangana, India), ciprofloxacin hydrochloride (Prati Donaduzzi, Toledo, Brazil), ethanol (Synth, Diadema, Brazil), acetic acid (99.7%, Synth, Diadema, Brazil), Mueller-Hinton broth (Difco, Sparks, NV, USA), Mueller-Hinton agar (Kasvi, Madrid, Spain), acetonitrile (>99.9%, Merck, Darmstadt, Germany), methanol (Biograde, Anápolis, Brazil), formic acid (≥95%, Synth, Diadema, Brazil), trifluoroacetic acid (≥99.5%, Sharlau, Barcelona, Spain), tetrabutylammonium sulphate (Êxodo Científica, Sumaré, Brazil), potassium phosphate (Dinâmica, Indaiatuba, Brazil), triethylamine (Êxodo Científica, Sumaré, Brazil), potassium hydroxide (Biotec, São José dos Pinhais, Brazil), orthophosphoric acid (Êxodo Científica, Sumaré, Brazil), dimethylsulfoxide (Synth, Diadema, Brazil), *Escherichia coli* ATCC 8739 (Lab-Elite™, St. Cloud, MN, USA)*, Enterococcus faecalis* ATCC 29212 (Lab-Elite™, St. Cloud, MN, USA)*, Staphylococcus aureus* ATCC 6538 (Lab-Elite™, St. Cloud, MN, USA)*, Staphylococcus epidermidis* ATCC 12228 (Lab-Elite™, St. Cloud, MN, USA)*, Pseudomonas aeruginosa* ATCC 9027 (Lab-Elite™, St. Cloud, MN, USA) and *Klebsiella pneumoniae* ATCC 10031 (Lab-Elite™, St. Cloud, MN, USA) were used in this study. All reagents were used as received.

### 3.2. Film Preparation Procedure

The films were prepared by dissolving chitosan at 1.5% *w/v* in 1% *v/v* acetic acid, while the crosslinker tannic acid (TA) and FeSO_4_ (Fe) were dissolved in ultrapure water at concentrations of 50 mg/mL and 3 mg/mL, respectively. The antibiotics were used at 5 mg/mL, dissolved in their respective solvents—ultrapure water for moxifloxacin and ciprofloxacin, 1% acetic acid for trimethoprim and ethanol 99% for sulfamethoxazole and linezolid.

Firstly, 2 mL of Fe, 0.48 mL of TA and 1.2 mL of the antibiotic were added to 8 mL of chitosan solution. The total volume was completed to 12 mL by the addition of ultrapure water. The final composition of the mixtures contained 20% TA, 5% FeSO_4_, and 5% of antibiotics related to the chitosan mass (100 mg). During the process, the mixture was stirred magnetically at 1500 rpm. Afterwards, 10 mL of the mixture was placed in a 9 cm diameter Petri dish and dried at 37 °C in an incubator, resulting in the films that were used for physicochemical characterization. The remaining mixture was placed into glass vials (277 µL in each vial) to prepare smaller samples for the release kinetics study. Chitosan films without antibiotic were also prepared.

Since the difference among the prepared films is the type of antibiotic, the films were named after the antibiotic acronyms, as shown in Table 7.

### 3.3. Antibiotic Release over Time

Antibiotic release over time was analyzed using 14 mm round films for each formulation, which were prepared inside glass vials, as reported in Section 3.2. To simulate physiological conditions, 1× PBS was used as a release medium. Firstly, two milliliters of PBS were added to each vial, and the medium was collected at 0 h, 1 h, 4 h, 24 h, 3 days, 7 days and every 7 days until the antibiotics were no longer quantifiable. After each sampling, fresh PBS was added. The collected media were stored at −20 °C for later quantification. These experiments were carried out in triplicates.

In order to investigate the antibiotic release mechanism, another experiment was carried out. In this experiment, collections were performed at shorter intervals (every fifteen minutes up to three hours), covering the initial release phase (60% of released drug) according to most drug release models, mainly those based on diffusion [46]. Fitting was performed for six different mathematical models: Korsmeyer-Peppas, Weibull, Zero order, First order, Hopfenberg and Higuchi, according to equations below, where *F* is the amount of drug released and t is the time.

Korsmeyer-Peppas, where *kKP* is the Korsmeyer-Peppas constant for incorporating structural modifications and geometric characteristics of the system and *n* is the diffusional exponent that indicates the transport mechanism:(1)$$F=kKP(t^{n})$$

Weibull, where *Ti* represents the release latency time, α represents the process time scale and β is the diffusional exponent that indicates the transport mechanism:(2)$$F=\{1-Exp[-({\left(t-Ti\right)}^{\beta })\alpha ]\}$$

Zero-order, where *K*0 is the apparent dissolution rate constant:(3)F=K0×t

First order, where *k*1 is the first order constant:(4)$$F=1-\exp\left(-k1\times t\right)$$

Hopfenberg, where *kHB* is the Hopfenberg constant and *n* is associated with the geometry of the matrix (film, sphere or cylinder):(5)$$F=[1-{\left(1-kHB\times t\right)}^{n}]$$

Higuchi, where *KH* is the Higuchi constant:(6)$$F=KH\times t^{0.5}$$

The fittings were evaluated by calculating the parameters R^2^, AIC (Akaike’s Information Criteria) and MSC (model selection criterion). The software Microsoft Excel DDSolver add-in and Origin^®^ were used to perform the fittings and the calculations [44].

### 3.4. Antibiotics Quantification

All antibiotics were quantified by high performance liquid chromatography (HPLC/UV) equipment (Model 20-S, Shimadzu, Kyoto, Japan). Aliquots were filtered through PTFE 0.22 µm, transferred to quantification vials, and analyzed according to the methods described below.

#### 3.4.1. Trimethoprim

Trimethoprim was quantified by HPLC/UV, using a, XTERRA RP18 column 150 mm × 4.6 mm with particle size 3.5 µm at 40 °C. Flow rate was 1.0 mL/min, injection volume 20 µL and wavelength detection 254 nm. The mobile phase included a buffer, acetonitrile (80:20), and the buffer was prepared with 794 mL of ultrapure water, 5 mL of glacial acetic acid and 1 mL of triethylamine [67].

#### 3.4.2. Moxifloxacin

Moxifloxacin was quantified by HPLC/UV, using an XTERRA RP18 column 150 mm × 4.6 mm with particle size 3.5 µm at 45 °C. Flow rate was 0.9 mL/min, injection volume 25 µL and wavelength detection 293 nm. The mobile phase included a buffer, acetonitrile (72:28), and the buffer was prepared dissolving 0.25 g of tetrabutylammonium hydrogen sulfate, 0.5 g of monobasic potassium phosphate and 1 mL of phosphoric acid in 1000 mL ultrapure water [68].

#### 3.4.3. Linezolid

Linezolid was quantified by HPLC/UV, using a Phenomenex C18 column 250 mm × 4.6 mm with particle size 5 µm at 40 °C. Flow rate was 1.0 mL/min, injection volume 40 µL and wavelength detection 254 nm. The mobile phase included a buffer—methanol (60:40), and the buffer was prepared by dissolving 6.8040 g of dibasic potassium phosphate and 2 mL of phosphoric acid (to adjust pH for 3.5) in 1000 mL ultrapure water. This method was developed based on different references to find the optimal conditions for analysis [69,70,71].

#### 3.4.4. Ciprofloxacin

Ciprofloxacin was quantified by HPLC/UV, using an XTERRA RP18 column 150 mm × 4.6 mm with particle size 3.5 µm at 30 °C. Flow rate was 1.0 mL/min, injection volume 15 µL and wavelength detection 278 nm. The mobile phase included a buffer, acetonitrile (87:13), and the buffer was prepared with phosphoric acid at 0.025 M and pH 3, adjusted with triethylamine [72].

#### 3.4.5. Sulfamethoxazole

Sulfamethoxazole was quantified by HPLC/UV, using an XTERRA RP8 column 150 mm × 4.6 mm with particle size 5 µm at 45 °C. Flow rate was 0.9 mL/min, injection volume 20 µL and wavelength detection 210 nm. The mobile phase included of a buffer, methanol (70:30), and the buffer was prepared by dissolving 6.8 g of potassium phosphate in 1 L of ultrapure water adjusted to pH 5.3 with potassium hydroxide solution at 20 g/L [73].

### 3.5. Films Characterization

#### 3.5.1. Thickness

A 90 mm diameter film of each formulation was used to measure thickness using a digital micrometer at five different positions. The results are presented as mean and standard deviation.

#### 3.5.2. Swelling

The films were cut into small squares of approximately 1 mg, which were immersed in two milliliters of 1× PBS (Phosphate Buffered Saline) and incubated for 24 h at 37 °C and 150 rpm. Afterwards, the wet samples were washed two times with ultrapure water, blotted with filter paper and weighed. The swelling degree was calculated according to Equation (7):(7)$$Sw=\;\frac{w_{s}-w_{0}}{w_{0}}\times 100\%$$
where *w*_0_ represents the initial weight and *w_s_* represents the swollen weight. The experiments were carried out in quintuplicates and the results are represented as the mean and standard deviation.

#### 3.5.3. Scanning Electron Microscopy

The morphology of the samples was investigated by scanning electron microscopy. The samples were deposited on a tape and then metallized with gold for 3 min. Subsequently, the analysis was carried out using a TESCAN VEGA3 microscope (TESCAN, Brno, Czech Republic), operating between 10 and 15 kV at a working distance of 7–8 mm; the images were taken at 500× magnification.

#### 3.5.4. Optical Phase Contrast Microscopy

Optical phase contrast microscopy was used to analyze the film’s morphology (Axio Imager 2 Pol Polarized Light Microscope, ZEISS, Göttingen, Germany). Films were prepared on a coverslip to flatten the samples and facilitate focus. Images were taken along a diagonal transect of the coverslips, with magnification of 20×.

### 3.6. Antibacterial Assays

#### 3.6.1. Bacteria Stock Preparation

Six bacteria, frequently found in BAI, were used in this study, *Escherichia coli*, *Enterococcus faecalis*, *Staphylococcus aureus*, *Staphylococcus epidermidis*, *Pseudomonas aeruginosa* and *Klebsiella pneumonia* [74]. Each bacterium was seeded on sterile Mueller-Hinton agar in Petri dishes and incubated in an inverted position overnight at 37 °C. A single colony was picked and placed in 20 mL of sterile Mueller-Hinton broth overnight at 37 °C and 150 rpm. Then, sterile glycerol at 20% was added and the bacteria aliquots were frozen at −20 °C.

#### 3.6.2. Film Antibacterial Activity

The disk diffusion, also known as the Kirby-Bauer test [75], was performed to check the antimicrobial activity of the samples against the tested bacteria. Approximately 1 × 10^8^ CFU/mL, quantified by log dilution enumeration method [76], were spread with a Drigalski spatula into 15 cm Petri dishes coated with fresh sterile Mueller-Hinton agar. Film samples (cut into 6 mm diameter disks) [77] were sterilized with UV irradiation at 254 nm for 15 min on each side, and then placed in Petri dishes containing the bacteria. Paper disks impregnated with antibiotic were used as positive control and paper discs without antibiotic were used as negative control. The dishes were incubated at 37 °C for 24 h in an inverted position. Afterwards, the inhibition zones were measured using a digital pachymeter in three different positions. Experiments were carried out in at least duplicates for each bacterium.

#### 3.6.3. Indirect Antibacterial Activity over Time

Films prepared in vials, previously sterilized, were used for this test. Two milliliters of sterile Mueller-Hinton broth was placed in each vial and the media was removed in 6 h, 24 h, 3 days, 7 days and then every 7 days. After each collection, fresh Mueller-Hinton broth was added to the vials. The collected aliquots were kept at −20 °C until analysis. Bacteria stock suspension of *E. coli* and *S. aureus* were thawed and diluted to a final concentration of c.a. 1 × 10^6^ CFU/mL in sterile Mueller-Hinton broth. Afterwards, 100 µL of the aliquots were added to 100 µL of bacteria suspension in a 96-well culture plate. Negative controls (blanks) were prepared by adding the inoculum to 100 µL of sterile Mueller-Hinton broth. Positive controls were prepared by adding the antibiotics at 10 µg/mL. The plates were incubated at 35 °C, 200 rpm for 8 h, or until OD_600_ reached 0.6–0.8. The bacterial survival was calculated using Equation (8). Experiments were carried out in triplicates.
(8)$$\%\;Bacterial\;survival=\frac{{sample\;OD}_{600}}{{blank\;OD}_{600}}\times 100\%$$

### 3.7. Biocompatibility

#### 3.7.1. Cell Culture

The effects of the films on cell viability were analyzed using Human Dermal Fibroblasts (HDFs, C0045C, Invitrogen Corporation, Burlington, ON, Canada). The cells were cultured in Dulbecco’s modified Eagle’s medium (D-MEM) with 10% foetal bovine serum (FBS), penicillin (100 U/mL) and streptomycin (100 U/mL), at 37 °C in a saturated atmosphere at 5% CO_2_. Culture medium was changed every 48 h until 85–90% of confluence was reached. Then, cells were enzymatically detached from the culture plates (0.05% trypsin, Gibco, Invitrogen Corporation, Burlington, ON, Canada) and reseeded at a ratio of 1:3 or used for experiments. Cells at passage 7 were used for the experiments.

#### 3.7.2. Indirect Cytotoxicity Test

An indirect cytotoxicity assay of the films was performed based on the ISO 10993-5:2009 procedure. All samples were previously sterilized by UV irradiation, undergoing two cycles of 15 min UV irradiation on each side. After that, samples were stored in sterile 24 multi-well plates until use. Briefly, 1 cm^2^ samples (n = 3 per time point) were immersed in 660 µL of D-MEM supplemented with 1% P/S and incubated at 37 °C in a saturated atmosphere at 5% CO_2_ for 1 day. After the incubation, samples of the different media were collected and used for the cytotoxicity test. Before putting them in contact with cells, the extracted media was supplemented with 10% FBS.

HDFs were seeded in the wells of 96 multi-well plates at a density of 20,000 cells/cm^2^ and incubated at 37 °C, 5% CO_2_ for 24 h in 100 µL/well of complete medium. The next day, the medium was removed and 100 µL of extract was added to the well containing the cells and incubated for 24 h. Normal HDF complete medium was used as a control. The extracts were then removed and 100 μL of 1× solution of resazurin sodium salt in complete medium was added to the cells and incubated for 4 h at 37 °C and 5% CO_2_. After the incubation, the solutions containing the now-reduced resorufin product were collected and fluorescence intensity at a 545 nm_ex_/590 nm_em_ wavelength was measured with a SpectraMax i3x Multi-Mode Plate Reader (Molecular Devices, San Jose, CA, USA). Fluorescence intensity is proportional to cell viability.

#### 3.7.3. Statistical Analysis

Statistical significance was calculated using ANOVA non-parametric Kruskal-Wallis method with Dunn post-test using the software InStat™. Values of *p* < 0.05 or less were considered significant.

## 4. Conclusions

Chitosan-based films crosslinked with tannic acid and iron were successfully loaded with five different antibiotics. The release behavior exhibited notable differences, categorized into three distinct groups: transient (linezolid and sulfamethoxazole), short-term (trimethoprim), and long-term (moxifloxacin and ciprofloxacin). The zwitterionic drugs showed the most sustained release kinetic profile compared to the positive, negative or neutral drugs. Thus, in addition to the well-known need for crosslinker selection, this work highlights the importance of drug structure selection for tuning drug release kinetics, which has not been reported in detail in the existing literature so far. Furthermore, the different releasing kinetics represent a tunable platform that, with rigorous criteria in terms of antibacterial application, can be applied in a variety of indwelling devices for short to sustained antibiotic release. In particular, the broad-spectrum antibiotics ciprofloxacin and moxifloxacin showed a prolonged release of about six months, indicating their potential application as coatings for medical devices requiring long indwelling times, such as central venous catheters. The future directions of this research will consist of developing methods of adhering films to different medical device substrates and characterization in terms of stability and biocompatibility in vitro and in vivo.

## Figures and Tables

**Figure 1 antibiotics-13-01055-f001:**
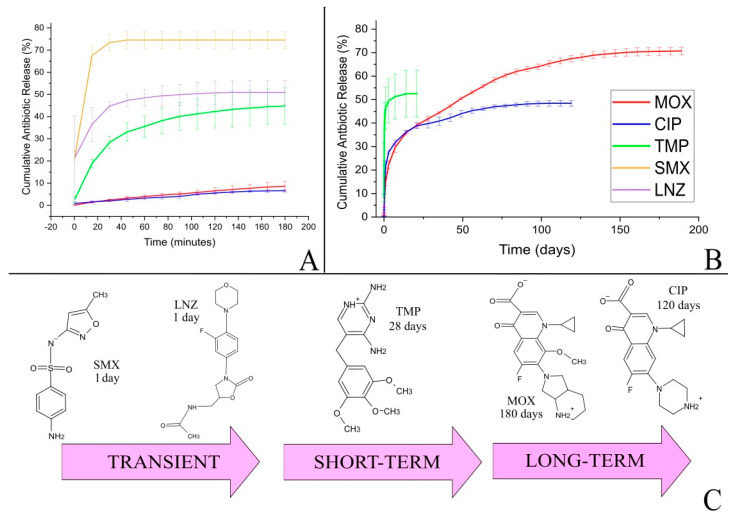
Cumulative antibiotic release from chitosan films crosslinked with tannic acid and iron until 200 min (**A**) and 200 days (**B**). (**C**) represents the antibiotic structure and release time. Studies carried out in PBS buffer at 37 °C, 100 rpm, pH 7.4.

**Figure 2 antibiotics-13-01055-f002:**
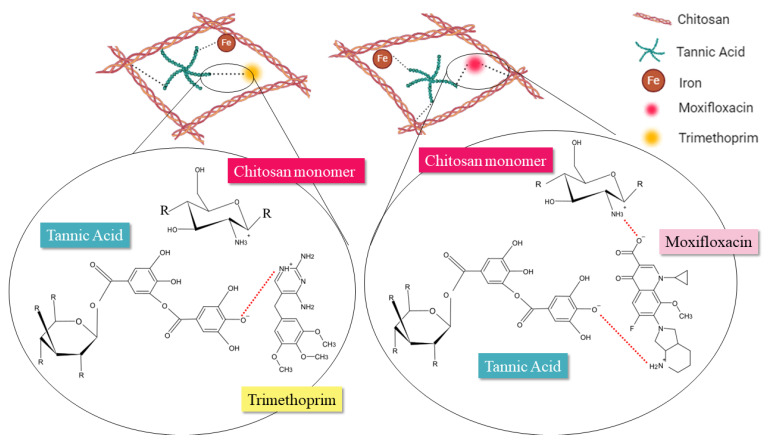
Schematic representation of intermolecular interactions between antibiotics and film matrix. Images produced with BioRender.

**Figure 3 antibiotics-13-01055-f003:**
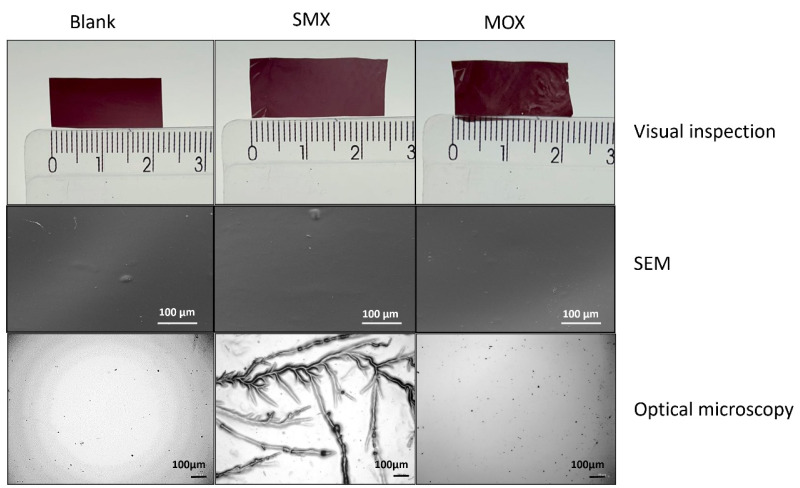
Visual inspection, optical microscopy and scanning electron microscopy analysis of blank film, sulfamethoxazole-loaded film and moxifloxacin-loaded film.

**Figure 4 antibiotics-13-01055-f004:**
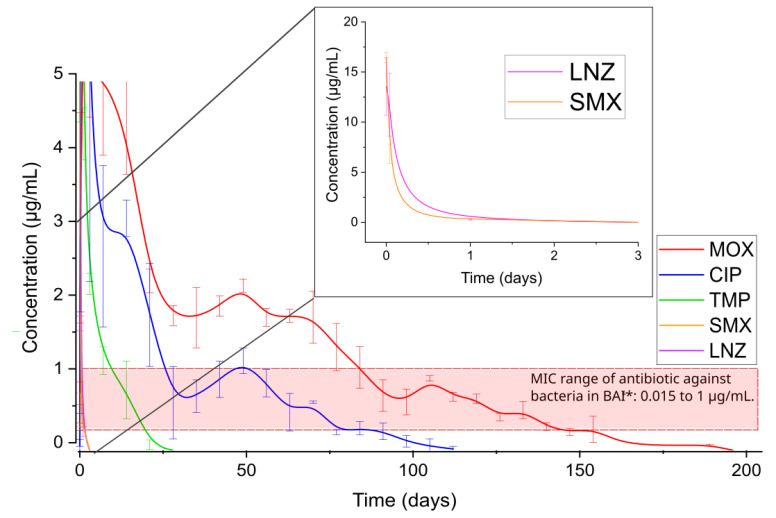
Antibiotic concentration released from chitosan-based films in PBS. The samples were made with chitosan, tannic acid, FeSO_4_ and the respective antibiotic listed on the label. * BAI: Biomaterial Associated Infection.

**Figure 5 antibiotics-13-01055-f005:**
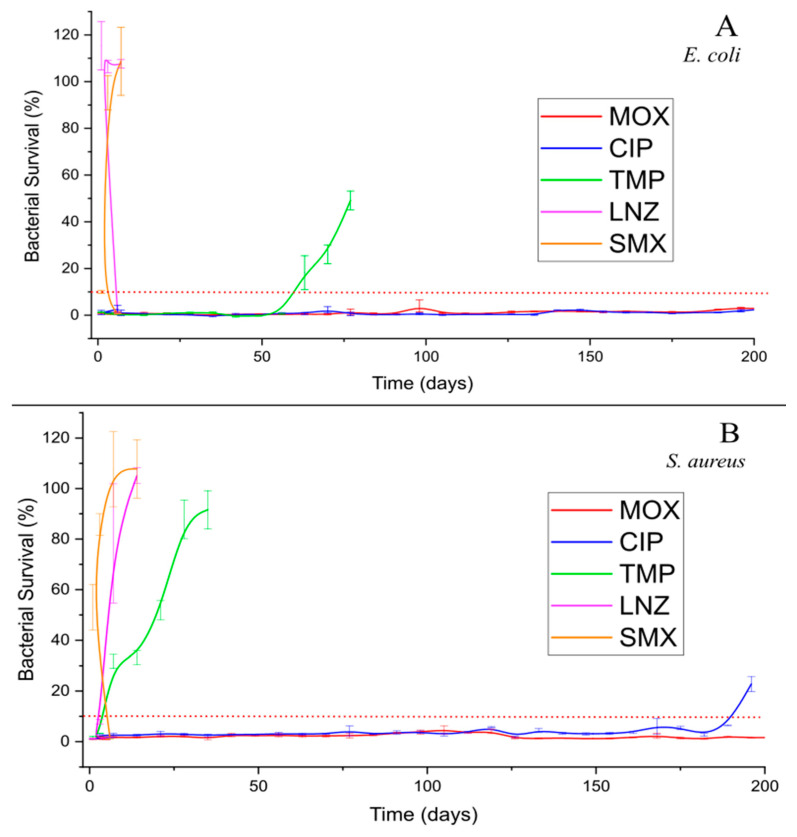
Indirect antibacterial activity over time of films against *E. coli* (**A**) and *S. aureus* (**B**). Experiments were performed in triplicates.

**Figure 6 antibiotics-13-01055-f006:**
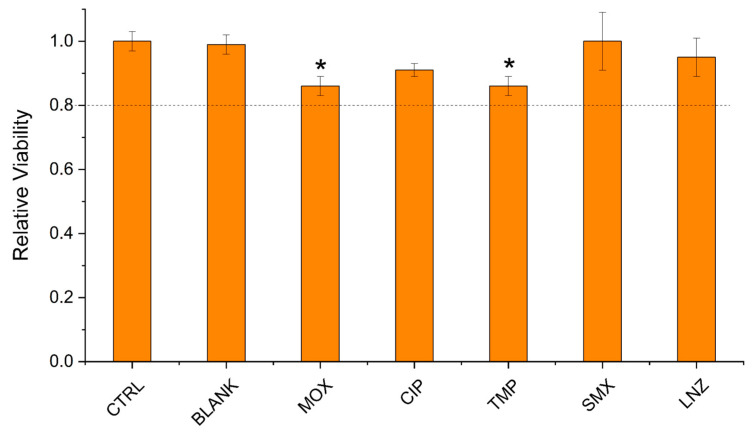
Indirect cytotoxicity test. Results obtained by treating HDFs with the extracts from the different experimental conditions. Cell viability was measured after 1 day of incubation by means of a resazurin salt solution. * Kruskal-Wallis method with Dunn post-test, *p* ˂ 0.05 vs. CTRL.

**Table 1 antibiotics-13-01055-t001:** Antimicrobial-coated PICC Central Venous Catheters available on the market.

Catheter	Brand	Impregnated Compound	Release Time	Ref.
Arrowg+ard^®^ Blue Plus	Arrow	Chlorhexidine and silver sulfadiazine	<30 days	[12]
Altius^®^ ProActiv+	Kimal	Polyhexamethylene biguanide	30 days	[13]
Multistar 3 UP	Vygon	Rifampicin + Miconazole	29 days	[14]
Chlorag+ard^®^	Arrow	Chlorhexidine	30 days	[15]

**Table 2 antibiotics-13-01055-t002:** Antibiotics molecular weight and charge at physiological pH.

Drug	MW (g/mol)	Charge	Water Solubility (mg/mL) **	Acronyms for Films ***
Moxifloxacin	401.2	+1 and −1 *	0.17	MOX
Ciprofloxacin	331.3	+1 and −1 *	1.35	CIP
Trimethoprim	290.3	+1	0.62	TMP
Sulfamethoxazole	253.0	−1	0.46	SMX
Linezolid	337.3	0	1.44	LNZ

* Zwitterionic compounds. ** Pubchem (Human Metabolomic Data Base). *** The films were named after the acronyms of the loaded antibiotic, since they share the same matrix composition.

**Table 3 antibiotics-13-01055-t003:** Release mechanism and goodness of fit (R^2^) for each antibiotic considering different mathematical models.

Model	Mechanism	MOX	CIP	TMP	SMX	LNZ
Korsmeyer-Peppas	*	**0.99**	**0.96**	**0.96**	**0.96**	**0.93**
Weibull	*	**0.99**	**0.97**	**0.99**	**0.86**	**0.92**
Higuchi	Diffusion	0.95	0.92	0.83	−0.97	0.91
Zero order	Dissolution	0.94	0.85	−0.30	−4.26	0.82
First order	Dissolution	0.95	0.86	0.35	−1.08	0.89
Hopfenberg	Erosion	0.94	0.86	0.35	−1.09	0.88

* Indicates different possible mechanisms such as diffusion, swelling, erosion or mixed. The best fittings are highlighted in bold.

**Table 4 antibiotics-13-01055-t004:** Kinetic parameters for the release mechanism interpretation.

Model	Parameter	MOX	CIP	TMP	SMX	LNZ
Korsmeyer-Peppas	**n** n < 0.50 = Fickian diffusion 0.50 < n < 1.0 = Anomalous (non-Fickian)	0.714	0.674	0.429	0.133	0.174
Weibull	**β** β < 0.35 = High disordered spaces 0.39 < β < 0.69 = Diffusion 0.69 < β < 1 = Diffusion + another mechanism	0.726	0.787	0.484	0.261	0.213
Release time		Long-term		Short-term	Transient	

**Table 5 antibiotics-13-01055-t005:** Thickness and swelling of chitosan-based films.

Film	Blank	MOX	CIP	TMP	SMX	LNZ
Thickness (µm)	21.0 ± 2.2	22.0 ± 1.9	23.0 ± 3.1	18.0 ± 1.8	21.0 ± 1.7	22.0 ± 2.4
Swelling (%)	247 ± 39	155 ± 37	172 ± 17	177 ± 33	221 ± 31	198 ± 23

**Table 6 antibiotics-13-01055-t006:** Disk diffusion of chitosan-based films loaded with different antibiotics against bacteria frequently found in BAI.

Films	*E. coli* (mm)	*K. pneumoniae* (mm)	*P. aeruginosa* (mm)	*S. epidermidis* (mm)	*S. aureus* (mm)	*E. faecalis* (mm)
MOX	21.7 ± 1.3	28.3 ± 0.9	19.4 ± 0.1	28.4 ± 2.6	27.5 ± 1.9	21.6 ± 0.6
CIP	29.5 ± 1.2	29.4 ± 0.0	33.7 ± 0.3	24.1 ± 0.5	24.4 ± 0.3	19.8 ± 0.5
TMP	25.3 ± 1.2	30.2 ± 3.0	0.0	32.7 ± 1.8	23.8 ± 0.1	31.1 ± 0.0
SMX	18.1 ± 0.6	15.8 ± 0.5	0.0	0.0	0.0	0.0
LNZ	0.0	18.5 ± 2.3	0.0	31.4 ± 2.8	26.9 ± 1.4	31.4 ± 0.0

**Table 7 antibiotics-13-01055-t007:** Film composition and acronyms.

Film Composition	Film Acronym
Chitosan, 20% tannic acid, 5% FeSO_4_ and 5% moxifloxacin	MOX
Chitosan, 20% tannic acid, 5% FeSO_4_ and 5% ciprofloxacin	CIP
Chitosan, 20% tannic acid, 5% FeSO_4_ and 5% trimethoprim	TMP
Chitosan, 20% tannic acid, 5% FeSO_4_ and 5% sulfamethoxazole	SMX
Chitosan, 20% tannic acid, 5% FeSO_4_ and 5% linezolid	LNZ

## Data Availability

Data are available in a publicly accessible repository. https://drive.google.com/drive/folders/1ofyftrweHNNDUS3P_lZvDzNLMURw4dWi (accessed on 8 October 2024).

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
