# Peer review of "From Burst to Sustained Release: The Effect of Antibiotic Structure Incorporated into Chitosan-Based Films"

_antibiotics, 2024, doi:10.3390/antibiotics13111055_

Round 1
Reviewer 1 Report
Comments and Suggestions for Authors
The authors have put quite some effort in writing the paper and overall it is well written. Here are some suggestions to further improve the paper:
1- In figure 2, and throughout the paper the authors suggested hypothesis on the formation of ionic interactions between drug and chitosan. Is there any way to confirm the hypothesis via analytical methods to showcase the formation of ionic interactions?
2- TMP and SMX had the lowest molecular weight and somehow they showed the fastest release. How important is the molecular weight of the antibiotic and its relation to the release rate? In terms of significance between molecular weight and functional groups which one is more important? It would be interesting to see some discussion on the topic to be added to the paper.
3- Figure 3, Why the so called grooves or vascular structure only happened for SMX sample? Afterall the preparation of the films followed the same protocol for all samples? The authors should explain.
4- Figure 6, MOX and TMP showed significant difference in relative viability in cytotoxicity. What is the significance of this results in terms of application in vivo for instance? It's a good idea to add some discussion on it.
5- Do authors have any data on different dosage of the crosslinkers added to chitosan to see the effect on the release rate? After all if the release rate can be tailored to match the concentration for the required antibacterial activity as in table 6 some show zero antibacterial properties (SMX and LNZ). Addition of this data can significantly improve the quality of the paper.
Author Response
Thank you very much for taking the time to review this manuscript. Please find the detailed responses below and the corresponding revisions/corrections have been done in the re-submitted files:
- In figure 2, and throughout the paper the authors suggested hypothesis on the formation of ionic interactions between drug and chitosan. Is there any way to confirm the hypothesis via analytical methods to showcase the formation of ionic interactions?
- Thank you for your question. To our best knowledge, there is no analytical method for characterizing ionic interactions in such a complex composition, since as soon as one of the components or the pH is modified, so are all the ionic interactions. However, these ionic interactions with chitosan are well known and described in literature (Cavallaro et al., 2021). Furthermore, this approach based on these ionic interactions are commonly used to build layer by layer coating (Hu, B. et al. 2021) using a cationic polymer such as chitosan and an anionic one containing carboxyl groups, e.g. hyaluronic acid, carboxymethylcellulose, etc.(Calais, G. B. et al. 2024, Calais, G. B. et al. 2023; Croll T. I. et al, 2006). We can therefore assume that chitosan amino groups can interact with moxifloxacin and ciprofloxacin carboxyl groups, enabling strong interactions of these antibiotics and the polymer matrix. Moreover, interactions between fluoroquinolones, such as moxifloxacin and ciprofloxacin, with catechol groups was also reported several times in the literature, and used for instance to adsorb antibiotics onto polydopamine substrates (Bai et al., 2023; Tan et al., 2017). Consequently, electrostatic interactions between the fluoroquinolones (moxifloxacin and ciprofloxacin) and tannic acid catechol groups are as well expected.
Refs.
Cavallaro, G. et al. (2021) Chitosan-based smart hybrid materials: a physico-chemical perspective. Journal of Materials Chemistry B, 3, 2021.
Hu, B. et al. (2021). Recent advances in chitosan-based layer-by-layer biomaterials and their biomedical applications. Carbohydrate Polymers, 271, 118427.
Calais, G. B. et al. (2024). Multivalent-copper-loaded layer-by-layer coating for antibacterial and instantaneous virucidal activity for protective textiles. Applied Surface Science, 676, 160945.
Calais, G. B. et al. (2023). Bioactive textile coatings for improved viral protection: A study of polypropylene masks coated with copper salt and organic antimicrobial agents. Applied Surface Science, 638, 158112.
Croll, T. I. et al. (2006). A blank slate? Layer-by-layer deposition of hyaluronic acid and chitosan onto various surfaces. Biomacromolecules, 7(5), 1610-1622.
Bai, H.; et al. (2023) Removal of fluoroquinolone antibiotics by adsorption of dopamine-modified biochar aerogel. Korean J. Chem. Eng., 40(1), 215-222
Tan, F. et al. (2017) Preparation of polydopamine coated graphene oxide/Fe3O4 imprinted nanoparticles for selective removal of fluoroquinolone antibiotics in water. Scientific Reports, 7: 5735.
2- TMP and SMX had the lowest molecular weight and somehow they showed the fastest release. How important is the molecular weight of the antibiotic and its relation to the release rate? In terms of significance between molecular weight and functional groups which one is more important? It would be interesting to see some discussion on the topic to be added to the paper.
- Thanks for asking, and indeed the MW could as well play a role in the release behavior. This was also considered, however no tendency was observed when comparing the antibiotics MW and release time, as shown in the table below:
Table 2. Antibiotics (from lowest to largest molecular weight), and their respective charges and release times.
Antibiotics |
SMX |
TMP |
CIP |
LNZ |
MOX |
MW (g/mol) |
253.0 |
290.3 |
331.3 |
337.3 |
401.2 |
Charge |
-1 |
+1 |
+1/-1 |
0 |
+1/-1 |
Release time |
1 day (Transient) |
28 days (Short-term) |
120 days (Long-term) |
1 day (Transient) |
180 days (Long-term) |
Although TPM and SMX have the lowest molecular weights, they were classified in different categories regarding the release time, i.e. TMP was considered to have short term release and SMX transient release. Moreover, when comparing CIP and LNZ, which have similar molecular weight, their release times were very different, 120 days for CIP and 1 day for LNZ.
If molecular weight played an important role in release time, the release mechanism would be controlled by swelling, which was not observed in the mathematical fitting. In fact, the fittings suggest that diffusion is the predominant mechanism. Therefore, the data lead to the conclusion that the molecular interactions between the antibiotics and the components of the films are more important than their molecular weight to determine the release rate.
For clarification for the readers, the following part was added in the paper (lines 136-147, p.4-5): “It was observed that antibiotics molecular weight had no influence on the release kinetics. Although trimethoprim and sulfamethoxazole have the lowest molecular weights, they were classified in different categories regarding the release time, i.e. trimethoprim was considered to have short term release and sulfamethoxazole transient release. More-over, when comparing ciprofloxacin and linezolid, which have similar molecular weights, their release times were very different, 120 days for ciprofloxacin and 1 day for linezolid. Additionally, information’s such partition coefficient octanol/water, topographic polar surface area, number of rotatable bonds which brings relation with structural properties of the antibiotics shows no clear trend with release, see supplementary Table S1. Therefore, the data lead to the conclusion that the molecular interactions between the antibiotics and the components of the films are more important than their molecular weight and structural properties to determine the release rate.”
3- Figure 3, Why the so called grooves or vascular structure only happened for SMX sample? Afterall the preparation of the films followed the same protocol for all samples? The authors should explain.
- The reviewer is right in pointing out that the groove structure was only observed in the SMX sample. This is indeed surprising since all the films were prepared according the same protocol; it seems to be a combination of different factors, such as solubility and intermolecular interactions of the antibiotic with the matrix. Moxifloxacin might interact more favorably with the chitosan and tannic acid. In contrast, sulfamethoxazole might not interact as well, leading to phase separation or aggregation, as the films dried slowly resulting in a different morphology. For clarity, water solubility of antibiotic was added in Table 2 (line 103, p.3). The text (lines 233-239, p. 7) was also modified in consequence in the manuscript: “On the other hand, optical microscopy revealed inner details of the film, indicating a significant heterogeneity in the morphology of SMX films while MOX and blank films showed good homogeneity. This phenomenon can be explained by the difference in solubility between SMX and MOX [52,53], and intermolecular interaction between the antibiotics and the polymer matrix within the films”.
4- Figure 6, MOX and TMP showed significant difference in relative viability in cytotoxicity. What is the significance of this results in terms of application in vivo for instance? It's a good idea to add some discussion on it.
- Thanks for the recommendation; it is a good suggestion indeed. Despite the differences in viability, all films are above the threshold recommended by the ISO 10993-5 standard. So, it was therefore added (lines 343-345, p.11): “This means that in terms of in vivo application, all the films are likely to be biocompatible, and it is highly motivating to continue the investigation on animals in future works.”
5- Do authors have any data on different dosage of the crosslinkers added to chitosan to see the effect on the release rate? After all if the release rate can be tailored to match the concentration for the required antibacterial activity as in table 6 some show zero antibacterial properties (SMX and LNZ). Addition of this data can significantly improve the quality of the paper.
- Thanks for the question. In fact, this project began by studying the dosage of tannic acid to control gentamicin for as long as possible. The best dosage found was 20% with a sustained release of the antibiotic for 3 days, this data has not been published, then it was discovered that iron (II) significantly prolonged the release of the antibiotic https://doi.org/10.3390/nano13030484. After these steps, the formulation was fixed to study the effect of the antibiotic, which was shown in the current manuscript. Based on our previous observations, it is possible to decrease the dose of tannic acid to release the antibiotic more rapidly, although the lack of activity observed in Table 6 is due to the lack of antibiotic activity against these strains.

Reviewer 2 Report
Comments and Suggestions for Authors
In this manuscript, N. F. Sczesny and colleagues present a detailed investigation on the effect of antibiotic structure in chitosan-based films, specifically focusing on drug release behavior. The study evaluates five different antibiotics with varying molecular weights and charges, characterizing their thickness, swelling, mass loss, morphology, antibiotic release profiles, and cytotoxicity.
The strength of this work lies in the thorough characterization of different antibiotics within the chitosan-based films, which reveals varying release profiles, ranging from transient to short-term and long-term. Notably, the manuscript reports long-term antibiotic release times that exceed previously documented studies. Overall, the manuscript is well-organized and relatively easy to follow. However, I have the following suggestions to improve the data presentation and overall readability:
1. The title of the article is focusing on antibiotic structures, have the authors considered including other aspects of the structures, such as hydrophilicity/hydrophobicity, polarity, 3D shapes or conformation, solubility and etc to be more comprehensive.
2. In page 2 of 19, line 84.
a. The reviewer recommends including references that support the intrinsic antibacterial properties of chitosan, as well as its dual antibacterial effect when combined with antibiotics.
b. Moreover, while the manuscript mentions combining chitosan and antibiotics, it's worth noting that many antibiotic combinations result in antagonistic effects. Is the combination of chitosan with these five specific antibiotics always synergistic? If so, please provide supporting data to substantiate this claim, or clarify whether further screening is needed to identify synergistic combinations.
3. In Figure 1, it seems that LNZ and CIP exhibit cumulative release profiles reaching only 40-50%. Is this typical for these antibiotics, or could there be an issue with the formulation or methodology? Clarification on whether this behavior is expected would be helpful for the readers.
4. The manuscript uses both full names and abbreviations of antibiotics inconsistently. The reviewer suggests standardizing the use of abbreviations throughout the manuscript for clarity and consistency.
5. In Figure 2, the reviewer suggests the authors to increase the font for better visualization.
6. In Table 3, the terms MSC and AIC are not explained sufficiently. It would be helpful to provide a brief explanation of what these abbreviations stand for and their significance for a general audience.
7. In Page 6 of 19, line 180, “as showed in Table 4” instead of “Table 5”.
8. In Table 6, what are the concentration used and any justifications?
9. In Figure 4, what is BAI stands for. Also, the reviewer suggests adding a zoom-in figure near 0-2 days for SMX and LNZ in supplementary figure.
10. In Figure 5, the reviewer asks the authors to make the error bar larger and bold and specify the number of replicates in the captions. Currently the error bar, for example the green curve, is not very clear.
11. The conclusion could be strengthened by expanding the discussion on how this work could impact the field, particularly in terms of potential clinical applications. Additionally, offering insights into the future direction of this research, such as optimizing the synergy between antibiotics and chitosan or extending the release time further, would provide a more forward-looking perspective.
Author Response
Thank you very much for taking the time to review this manuscript. Please find the detailed responses below and the corresponding revisions/corrections have been done in the re-submitted files:
- The title of the article is focusing on antibiotic structures, have the authors considered including other aspects of the structures, such as hydrophilicity/hydrophobicity, polarity, 3D shapes or conformation, solubility and etc to be more comprehensive.
- The parameters related to the number of heavy atoms, Csp3, logP, TPSA, number of rotatable bonds were calculated with swiss param online tool. There was no correlation between the release kinetics and these physicochemical information of the antibiotic structure. The following text has been added to the manuscript and a table with calculated parameters has been added to the Supplementary Material.
Line 142-145, p 4“Additionally, information’s such partition coefficient octanol/water, topographic polar surface area, number of rotatable bonds which brings relation with structural properties of the antibiotics shows no clear trend with release, see supplementary Table S1.”
- In page 2 of 19, line 84.
- The reviewer recommends including references that support the intrinsic antibacterial properties of chitosan, as well as its dual antibacterial effect when combined with antibiotics.
- Thanks for the recommendation. Additional references were added (lines 85-87, p.2), as follows: “Moreover, chitosan has intrinsic antibacterial properties [20,21] and, together with the loaded antibiotic, it can have a dual antibacterial effect, contact killing and drug releasing.”
- Kong, M.; Chen, X.G.; Xing, K.; Park, H.J. Antimicrobial Properties of Chitosan and Mode of Action: A State of the Art Review. Int. J. Food Microbiol. 2010, 144, 51–63, doi:10.1016/j.ijfoodmicro.2010.09.012.
- Akincibay, H.; Åženel, S.; Ay, Z.Y. Application of Chitosan Gel in the Treatment of Chronic Periodontitis. J. Biomed. Mater. Res. - Part B Appl. Biomater. 2007, 80, 290–296, doi:10.1002/jbm.b.30596.
- Moreover, while the manuscript mentions combining chitosan and antibiotics, it's worth noting that many antibiotic combinations result in antagonistic effects. Is the combination of chitosan with these five specific antibiotics always synergistic? If so, please provide supporting data to substantiate this claim, or clarify whether further screening is needed to identify synergistic combinations.
- The reviewer raises an interesting point. In this work, we focus on antibiotic release and the data collected, including antibacterial assays, bearing on this point. On the other hand, chitosan is a material well known for its antibacterial properties, particularly on contact, so this is a hypothesis of two possible phenomena since chitosan is trapped in the film. To clarify matters for readers, the following text has been added (lines 87-89, p.2): “Furthermore, in some cases synergic effects with antibiotics were observed [22], where antibiotic potency increased with combination. However, in this work, the focus was on antibiotic release, and synergistic effects were not evaluated”.
- In Figure 1, it seems that LNZ and CIP exhibit cumulative release profiles reaching only 40-50%. Is this typical for these antibiotics, or could there be an issue with the formulation or methodology? Clarification on whether this behavior is expected would be helpful for the readers.
- Thanks for the comment. The total release percentage can be a result of the combination of different factors, especially the antibiotic electric charge, which influences the intermolecular interactions, and water solubility, which influences the distribution of the antibiotic molecules inside the polymeric film. According to literature data, the water solubility data of the five antibiotics were compiled and compared to the release percentages and electric charges.
Table 2. Antibiotics solubility in water compared to their respective release percentages and electric charges.
Antibiotic |
MOX |
SMX |
TMP |
CIP |
LNZ |
Water solubility (mg/mL)* |
0.168 |
0.459 |
0.615 |
1.35 |
1.44 |
Release percentage (%) |
70% 180 days |
75% 1 day |
50% 28 days |
45% 120 days |
50% 1 day |
Charge |
+1/-1 |
-1 |
+1 |
+1/-1 |
0 |
*Pubchem HMDB
The antibiotics with lower solubility (MOX and SMX) are released at higher percentages, while the antibiotics with higher solubility (TMP, CIP and LNZ) are released at lower percentages. This can be explained by the fact that the least water-soluble antibiotics are not well dispersed in the polymer matrix, form aggregates and tend to be distributed in the polymer surface (Berkland et al., Microsphere size, precipitation kinetics and drug distribution control drug release from biodegradable polyanhydride microspheres. Journal of Controlled Release 94 (2004) 129 – 141). By opposite to MOX, MOX and SMX interact strongly with the polymer matrix, what delays its release. On the other hand, TMP, CIP and LNZ are more water-soluble and lead to a better dispersion in the polymer matrix, what imparts the release of all the incorporated antibiotic, which remains at a maximum of 50% release.
To support this discussion and to clarify these behaviors for the readers, the following section was added (lines 169-181, p. 5): “It is also possible to notice that the antibiotics exhibit different release percentages, which may be due to different factors or their combinations: such as the antibiotic electric charge, which influences the intermolecular interactions, and water solubility, which influences the distribution of the antibiotic molecules in the polymeric film. These main factors are summarized in Table 2. For instance, it can be seen that antibiotics with lower solubility (MOX and SMX) were released at higher percentages, while the antibiotics with higher solubility (TMP, CIP and LNZ) were released at lower percentages (Figure 1). This may be explained by the fact that the less water-soluble antibiotics are not well dispersed in the polymer matrix, may form aggregates and tend to be distributed over the polymer surface [44]. The difference between MOX and SMX is that MOX interacts strongly with the polymer matrix, thus delaying its release. TMP, CIP and LNZ, on the other hand, are more water-soluble and tend to disperse in the polymer matrix, so that the release of all incorporated antibiotics remains at a maxi-mum of 50%.”
- The manuscript uses both full names and abbreviations of antibiotics inconsistently. The reviewer suggests standardizing the use of abbreviations throughout the manuscript for clarity and consistency.
- The reviewer is right using both full names and abbreviations could become confusing. Therefore, the manuscript was standardized and corrections made in this sense. The acronyms are now used only to indicate the formulations of the films containing different antibiotics.
- In Figure 2, the reviewer suggests the authors to increase the font for better visualization.
- Thanks to the reviewer for his suggestion, and the font size was increased for better visualization in the resubmitted version of the manuscript.
- In Table 3, the terms MSC and AIC are not explained sufficiently. It would be helpful to provide a brief explanation of what these abbreviations stand for and their significance for a general audience.
- According to the reviewer suggestion, an explanation for these terms was included.
- In Page 6 of 19, line 180, “as showed in Table 4” instead of “Table 5”.
- Thank you for the comment, the correction was made.
- In Table 6, what are the concentration used and any justifications?
- A comment about the antibiotic dose in the film discs used for the antimicrobial analysis was added in the text after the table, as follows (lines 280-284, p.9): “The disk diffusion showed that the films incorporated with the antibiotics possesses strong antibacterial properties, considering that all the antibiotics were completely incorporated after the casting, the dose of antibiotics in the 6 mm disc was 22 µg, which is higher than the dose for standard sensibility disc-diffusion tests [60], and therefore anti-bacterial activity is expected upon antibiotic release.”
- In Figure 4, what is BAI stands for. Also, the reviewer suggests adding a zoom-in figure near 0-2 days for SMX and LNZ in supplementary figure.
- Figure 4 was corrected according to the suggestions.
- In Figure 5, the reviewer asks the authors to make the error bar larger and bold and specify the number of replicates in the captions. Currently the error bar, for example the green curve, is not very clear.
- Figure 5 was corrected according to the suggestions.
- The conclusion could be strengthened by expanding the discussion on how this work could impact the field, particularly in terms of potential clinical applications. Additionally, offering insights into the future direction of this research, such as optimizing the synergy between antibiotics and chitosan or extending the release time further, would provide a more forward-looking perspective.
- As suggested by the reviewer, the conclusion was modified and developed as follows (lines 591-601, p.16):” Thus, in addition to the well-known need for crosslinker selection, this work highlights the importance of drug structure selection for tuning drug release kinetics, which was not reported in details in the existing literature so far. Furthermore, the different releasing kinetics represents a tunable platform that with rigorous criteria in terms of antibacterial application can be applied in a variety of indwelling devices from short to sustained anti-biotic release. In particular, the broad-spectrum antibiotics ciprofloxacin and moxifloxacin showed a prolonged release of about six months, indicating their potential application as coatings for medical devices requiring long indwelling times, such as central venous catheters. The future directions of this research will consist in the development of adherence methods of the films on different medical devices substrates and characterization in terms of stability and biocompatibility in vitro and in vivo.”

Reviewer 3 Report
Comments and Suggestions for Authors
Review of the Manuscript ANTIBIOTICS 3273231 entitled “From burst to sustained release: the effect of antibiotic structure incorporated into chitosan-based-films” to be considered for publication in the JOURNAL OF ANTIBIOTICS (MDPI).
In this paper, authors report their findings on the antibacterial potential of chitosan-based films with five antibiotics (moxifloxacin, ciprofloxacin, trimethoprim, linezolid, and sulfamethoxazole). Samples were characterized using SEM, optical microscopy, liquid chromatography, and tests for antibacterial activity and antibiotic release. Authors report that depending on a careful selection of the antibiotic structure will influence the antibiotic release kinetics to be specific to medical devices.
This is a very interesting and cumbersome research with valuable results worthy of publication.
Reviewer suggestion: accept
Author Response
Thank you very much for taking the time to review this manuscript.
Reviewer 4 Report
Comments and Suggestions for Authors
In this study, the authors present a comprehensive investigation of the full profiles of five different antibiotics incorporated into a previously developed chitosan-based biofilm. Their research includes the evaluation of release kinetics and antimicrobial activity over various timeframes. Importantly, no cellular toxicity was observed, suggesting a broad spectrum of safety for the chitosan-based biofilm. Notably, the five antibiotics exhibited significantly different release profiles and antimicrobial activities, which the authors classified as transient, short-term, and long-term. The mechanistic analysis further supports the significance of this research, strengthening its potential for publication. Overall, this is an interesting and valuable study, requiring only minor revisions before publication.
-
Several additional key characterizations of the biofilm are recommended for a more complete evaluation: a. Drug loading efficiency and content for the five antibiotics used in the study. b. Mechanical properties of the biofilms, such as stiffness and flow behavior. c. Stability of the antibiotics within the biofilm under various conditions and timeframes. While this aspect is missing, it should not prevent publication.
-
Including a more detailed background and rationale for selecting these five antibiotics in the biofilm would enhance the study’s scientific impact. This should also relate to the authors' central argument and discuss potential future applications.
-
The discussion section is currently missing and should be added to the manuscript.
-
Mass loss data is absent in Table 5 and should be included.
-
In Figure 6, the control lacks standard deviation. Additionally, the statistical analysis should be clearly defined in the figure legend and appropriately labeled in the figure.
Author Response
Thank you very much for taking the time to review this manuscript. Please find the detailed responses below and the corresponding revisions/corrections have been done in the re-submitted files.
- Several additional key characterizations of the biofilm are recommended for a more complete evaluation: a. Drug loading efficiency and content for the five antibiotics used in the study. b. Mechanical properties of the biofilms, such as stiffness and flow behavior. c. Stability of the antibiotics within the biofilm under various conditions and timeframes. While this aspect is missing, it should not prevent publication.
- Thank you for the suggestions. Regarding the mentioned topics:
- Drug loading was not measured because the chitosan films containing crosslinkers and antibiotics are prepared by casting, meaning that all added antibiotic is present in the formulations.
- Mechanical properties of the films will be measured in the future steps of this research, which will aim at developing coatings for central venous catheters.
- Stability of the antibiotics within the films will also be measured in the future, in which the behavior of the coatings under simulated blood flow will be analyzed.
- Including a more detailed background and rationale for selecting these five antibiotics in the biofilm would enhance the study’s scientific impact. This should also relate to the authors' central argument and discuss potential future applications.
- We rephrase the paragraph to “In this study, five different antibiotics were selected based on their molecular weight to investigate the effects of the size on release kinetics, solubility to observe the if there is a trend on low solubility and slow release, and diversity of functional groups and electrostatic charge to relate with the intermolecular interaction between the antibiotics and the films matrix (polymer and crosslinkers interactions), as shown in Table 2.” (lines 98-102, p.3)
- The discussion section is currently missing and should be added to the manuscript.
- The discussion was written along with the results in the section Results and Discussion.
- Mass loss data is absent in Table 5 and should be included.
- Mass loss data was removed from the Methods and Results and Discussion sections because it did not elucidate the release behavior. All obtained films showed an average of 20% mass loss without statically significant differences. Since the release mechanism was shown to be diffusion-controlled, these data were not considered relevant.
- In Figure 6, the control lacks standard deviation. Additionally, the statistical analysis should be clearly defined in the figure legend and appropriately labeled in the figure.
- Thanks for the comments. The suggestions were implemented.

Round 2
Reviewer 1 Report
Comments and Suggestions for Authors
The paper has been revised and improved. It is easier to follow and clearer at current state.
Reviewer 2 Report
Comments and Suggestions for Authors
The reviewer appreciates the authors' efforts in addressing the comments and implementing the suggested revisions. I am pleased to see that the revisions have improved the clarity and overall flow of the work, and I think the manuscript is now suitable for publication.